# Automated Analysis of Conservation Strategies for Endangered Species through AI-Driven Literature Review

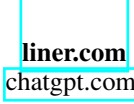

**liner.com**
chatgpt.com

## Abstract

In the era of the 'Great Extinction' driven by climate change and environmental pollution, developing effective and efficient strategies for the conservation of endangered species has become a critical global challenge. Recent advances in artificial intelligence (AI) have enabled the automation of analysis of academic literature and ecological monitoring, allowing researchers to allocate more resources to on-the-ground conservation efforts. This study reviews recent applications of AI-based systems in literature review, ecological monitoring, and risk detection, and compares these approaches with traditional conservation methods. Based on this analysis, the study proposes future directions for technological innovation and offers policy recommendations to enhance the effectiveness of biodiversity conservation.

## 1 Introduction

The loss of biodiversity has been exacerbated worldwide due to habitat changes and the invasion of alien species, underscoring the growing need for research on the conservation of endangered species.

### 1.1 AI applications in Biodiversity loss studies

In today's academic environment, where an overwhelming number of research articles are published daily, it is virtually impossible for a single researcher to manually review all relevant literature. Artificial intelligence (AI) has emerged as a key tool to overcome this limitation. AI and automated analytical technologies have enhanced species monitoring and habitat assessment, thus improving the efficiency and precision of conservation efforts. In addition, AI-based literature analysis systems can rapidly and accurately process vast amounts of data, extracting and summarizing key information, reducing the time researchers must spend on analysis. Several studies have shown that these systems outperform conventional methods in improving the accuracy of the summarization of scientific knowledge. Such technologies allow researchers to save time in reviewing literature and dedicate more resources and effort to developing and implementing strategies in conservation practice.

### 1.2 Purpose of Literature review

The purpose of this review of the literature is to comprehensively analyze the urgency of protecting endangered species, the limitations of existing strategies, the necessity of AI and automated analysis, and the expected effects of these technologies. This review addresses the impact of AI on biodiversity monitoring and decision-making support, based on the latest literature and systematic review.

## 2 Method of Collecting and Analyzing literature

This review gathered studies on the conservation of endangered species from major databases such as Web of Nature, Science, Scopus, and Google Scholar. Search queries included "Endangered species

conservation strategies," "Applications in wildlife conservation," "Species-specific conservation," "Conservation across regions," and more.

Clear inclusion and exclusion criteria were applied in the selection of studies, considering research topics, periods, languages, and data accessibility. Papers include those that specified species and strategies and provided indicators related to conservation effectiveness. Simple ecological descriptions or policy reports without data were excluded. The final set of studies were classified based on criteria such as species classification and strategy type.

# 3 Research trends by Conservation strategy type

## 3.1 Habitat conservation

Habitat conservation is a core strategy for protecting endangered species, focusing on habitat protection to ensure species survival (Figure 1 a, Supplementary Table 1). A representative example is the Giant Amazon River Turtle Protection Program in the Amazon rainforest, which has achieved population recovery and reduced poaching over 40 years of effort and community involvement (Figure 1 b). Well-managed protected areas have experienced fewer disturbances than nearby unprotected areas and contributed to the protection of wildlife, water resources, and forests. However, limitations exist, such as the influx of invasive species and limited habitat range, and consistent effectiveness may not be observed across all species.

## 3.2 Species restoration

Species restoration is an important conservation tool that restores biological communities and ecosystems by reintroducing locally extinct species (Figure 1 a, Supplementary Table 1). Successful cases include the reintroduction of red deer in Portugal and European badgers, showing long-term population growth and habitat expansion (Figure 1 b). However, species with complex social structures, high intelligence, or those requiring extensive training have lower success rates, and further research is needed on maintaining genetic diversity and the effects of reintroduction for long-term success.

## 3.3 Policies and Laws

Policies and laws for endangered species conservation provide essential foundations for biodiversity protection(Figure. 1a, Supplementary Table 1). The U.S. Endangered Species Act (ESA) specified species protection through habitat conservation, recovery plans, and international trade regulations, while CITES has regulated international trade of endangered species since 1975. These laws have contributed to reducing biodiversity loss, but their effectiveness has varied due to implementation issues and jurisdictional conflicts.

## 3.4 Community participation

Community participation-based conservation strategies are essential components for sustainable conservation efforts(Figure. 1a, Supplementary Table 1). According to research, the involvement of local residents has led to positive ecological and social outcomes. The CBC approach implemented by the pastoral community in the Loliondo region of northern Tanzania improved community livelihoods and contributed to an increase in wildlife species, water resource protection, and forest area(Figure. 1b). A 40-year CBCM program in the Brazilian Amazon successfully led to the recovery of the Giant Amazon River Turtle population and reduced poaching activities to 2 percent.

## 3.5 Monitoring and Data management

Monitoring and data management strategies for endangered species conservation include remote sensing, IoT, genetic analysis, and AI technologies(Figure. 1, Supplementary Table 1).

**Remote sensing applications** Remote sensing utilizes satellite imagery and drones to enable habitat mapping, species distribution monitoring, and illegal activity detection. Modeling vegetation structure using LiDAR information is useful for species occurrence, abundance, and community modeling,

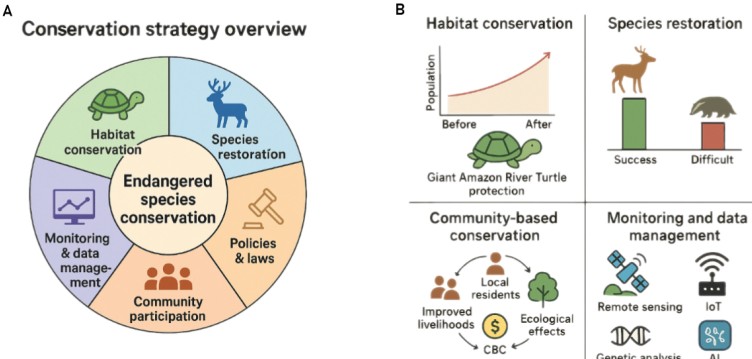

Figure 1: **Conservation strategies for endangered species. (A)** Overview of major strategies for endangered species conservation, including habitat conservation, species restoration, monitoring and data management, community participation, and the establishment of policies and laws. **(B)** Representative examples of each strategy. Habitat conservation is illustrated by the Giant Amazon River Turtle protection project, showing population recovery after intervention. Species restoration depicts contrasting outcomes, with some species showing successful recovery while others remain difficult to restore. Community-based conservation (CBC) emphasizes the role of local residents, leading to improved livelihoods and ecological benefits through reciprocal interactions. Monitoring and data management highlights the use of advanced technologies such as remote sensing, IoT, genetic analysis, and AI to enhance conservation efforts.

and is cost-effective at about 5 percent of field survey costs. These methods are particularly used for efficiently monitoring wildlife populations in hard-to-access areas.

**IoT applications**    IoT is used to collect real-time data from wildlife habitats through a network of sensors, GPS trackers, and AI-powered cameras. These devices generate large amounts of data on animal movement, behavior, and environmental variables, providing insights needed for anti-poaching and habitat conservation. Specifically, IoT systems integrated with drones contribute to biodiversity monitoring and disease control in remote areas.

**Genetic analysis applications**    Genetic analysis is essential for assessing genetic diversity, analyzing population structure, monitoring genetic health, and identifying species. Next-generation sequencing (NGS), DNA barcoding, and environmental DNA (eDNA) analysis are emphasized, allowing for the collection of large amounts of genetic data non-invasively.

**AI applications**    AI is used for species identification, habitat suitability modeling, anti-poaching, automated monitoring, data analysis, and predictive modeling. In particular, computer vision and machine learning are used to analyze camera trap images and acoustic data to automatically identify species, track population trends, and detect illegal activities. AI-based predictive models analyze environmental data and species behavior patterns to predict habitat changes and optimize resource allocation, guiding proactive conservation measures.

## 4    Species and Region-specific strategy application differences

### 4.1    Analysis by species group

Endangered species conservation strategies show specific characteristics depending on the species group, such as mammals, birds, and plants (Figure. 2a). In mammal conservation, maintaining genetic diversity, integrating breeding techniques, and monitoring populations are critically addressed. Large mammals, in particular, focus on anti-poaching and preventing habitat destruction. For bats, research

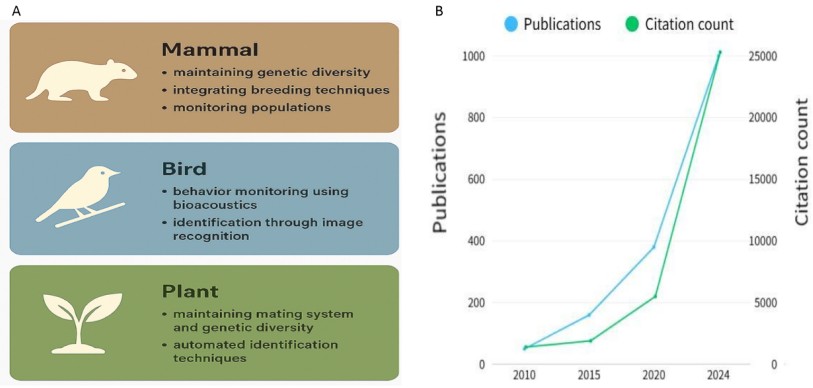

Figure 2: **Multi-faceted Analysis of Endangered Species Conservation Strategies (A)** Overview of the differences in conservation strategies applied to mammals, birds, and plants. **(B)** Graph showing the number of publications and citations in the field of environmental monitoring and data analysis using AI and ML. The number of publications has steadily increased since 2010, with a rapid surge after 2020. On average, the number of publications increased by 28.1 percent per year, while the number of citations grew by 35.6 percent per year.

suggests a need for a separate risk assessment framework due to their unique vulnerability to pesticide exposure, caused by high energy demands from flight and fat-soluble pesticide accumulation in fat stores. Bird conservation focuses on population and behavior monitoring using bio-acoustics, species identification through image recognition, and securing migration routes and habitat connectivity. Birds are vulnerable to habitat fragmentation, and adaptation and mitigation strategies for climate change are crucial. Plant species conservation focuses on maintaining mating systems and genetic diversity. Automated identification techniques can enhance endangered plant population monitoring and regional biodiversity research.

## 4.2 Differences in strategy application by continent and country

Endangered species conservation strategies exhibit different characteristics by continent and country (Figure. 2b). China, India, and Southeast Asia focus on establishing protected areas and restoring habitats for endangered plants and large mammals (e.g., Amur leopard, rhinoceros, tiger, Asian elephant). In North America, long-term land use planning is emphasized to secure habitat connectivity, along with incentives for private landowners' conservation participation and legal protection efforts. Adaptation strategies to reduce birds' vulnerability to habitat changes due to climate change are also significantly addressed. In Africa, in-situ conservation in politically unstable regions, active intervention for specific species protection, and community-based conservation approaches are emphasized.

## 4.3 Strategy selection trends by threat factor

Conservation strategies are tailored to specific threat factors such as habitat loss, climate change, invasive species, and over-exploitation. Habitat loss and fragmentation are primarily caused by land clearing, urbanization, and agricultural expansion, and corresponding strategies include habitat conservation and restoration, protected area establishment, and maintaining habitat connectivity. Climate change significantly impacts species distribution and habitat, with low-carbon energy sources being discussed as a solution. Invasive species management plays a crucial role in reducing negative impacts on ecosystems and affects the success of reintroducing specific species. To prevent over-exploitation and poaching, drone-based monitoring and AI-powered real-time warning systems are being introduced.

# 5   Advancement of AI technology-based conservation strategies

## 5.1   Cases of Surveillance and Monitoring

AI technology is widely applied in surveillance systems, population tracking, and illegal poaching prevention for the conservation of endangered species. Drone LiDAR is used in research to improve the accuracy of tree species classification by combining LiDAR and hyperspectral sensors, and the National Institute of Forest Research uses Drone LiDAR to search for endangered species in mountainous areas or cliffs that are difficult to access. The University of Bath in the United Kingdom succeeded in accurately calculating the African elephant population in a short time by combining a high-resolution satellite image (WorldView-3) and a deep learning algorithm. This is the first reliable monitoring of large-scale entities in hard-to-reach areas, demonstrating the potential of satellite remote sensing and AI-based analytics. In addition, in Tanzania's Serengeti National Park, using AI to analyze drone footage data to identify poachers has led to a more than 50 percent reduction in illegal hunting. The National Institute of Biological Resources in Korea successfully identified 27 species of Amazon parrots, which were morphologically difficult to distinguish, using a deep learning-based object recognition model, achieving 100 percent accuracy for the yellow-billed Amazon and blue-cheeked Amazon species. Additionally, Google AI's Perch model analyzes massive amounts of acoustic data to track populations of birds, mammals, amphibians, and marine life. In Hawaii, this technology was used to detect calls from honey-eaters 50 times faster than conventional methods, allowing for wider monitoring.

## 5.2   Cases of Risk detection and prevention

The IUCN and Huawei Tech 4Nature partnership uses artificial intelligence and blockchain to monitor potential threats to biodiversity on land and in the ocean. The 'Smart Buoy' detects the calls of marine mammals and transmits them to the cloud to evaluate behavior patterns and the effects of noise pollution through AI analysis. Another example, Nature Guardian, which detects threatening sounds like chainsaws or gunshots and immediately sends an alert to managers. In China, robots that combine multi-sensors and AI visual systems are deployed at the site to monitor illegal activities. African Parks uses BatHawk drones to proactively block poaching. In addition, Jellyfish-Bot, which mimics the movement of jellyfish, is contributing to the conservation of underwater ecosystems by collecting marine litter.

## 5.3   Advantages

Technology-based strategies increase efficiency through automated monitoring and data analysis, which is more accurate than the traditional method, allowing large areas to be monitored with less manpower, and real-time Through data collection and analysis, the accuracy and speed of conservation decisions are improved. In addition, AI is a powerful tool in the conservation of endangered species and is effective when combined complementarily with traditional conservation technologies and biological methods such as genetic analysis and artificial insemination. Using AI technology in harmony with traditional knowledge along with data sharing platforms, international cooperation, and policy support, it is possible to simultaneously achieve habitat protection and population recovery for endangered species and establish sustainable and safe conservation strategies.

## 5.4   Limitations

Technology-based conservation systems can require high initial installation costs and involve challenges such as data bias, ethical issues, and imbalances in data accessibility. Especially in developing countries, funding for technology adoption, maintenance costs, and lack of technical capabilities pose significant limitations.

# 6   Comprehensive Analysis

**Comparison of Strategy Frequency and Effectiveness**   Among endangered species conservation strategies, AI, machine learning (ML) and deep learning (DL) based monitoring and species identification, along with genetic diversity assessment and genetic analysis, were most frequently

mentioned, followed by habitat conservation and restoration. This suggests that while traditional habitat conservation remains important, the integration of advanced technologies is rapidly expanding. AI-based monitoring shows high effectiveness in species identification, behavior pattern analysis, and anti-poaching. Genetic analysis provides essential information for the success of population management and restoration programs

**Trend of Change Over Time**   Over the past five years, there has been a significant increase in the use of AI, ML, and DL-based monitoring and identification strategies. These technologies have proven effective in analyzing large datasets, predicting ecological trends, and recommending targeted conservation strategies. Notably, much of the focus has been on automating species identification using computer vision and machine learning, which has significantly improved the efficiency of population monitoring. Additionally, the use of drone technology for wildlife anti-poaching efforts has seen considerable growth.

**Changes in Technology Utilization Rate**   The integration of AI, IoT, drones, and genomic technologies in wildlife conservation is rapidly increasing. AI-based methods automate species identification and habitat monitoring, while drones enhance real-time surveillance and targeted intervention. These technologies significantly enhance the efficiency and accuracy of endangered species conservation efforts.

**Role of Endangered Species Conservation Using AI Technology**   AI technology plays a pivotal role in the conservation of endangered species by harnessing diverse data sources, including climate change patterns, pollution levels, and animal movement behaviors. This enables real-time analysis, empowering conservation managers to swiftly detect and respond to emerging crises. Machine learning algorithms automate key tasks such as species identification, population monitoring, and habitat mapping, thereby facilitating the development of more efficient conservation strategies with reduced human intervention. Additionally, the automation of data collection and analysis enhances both the accuracy and speed of research, cutting down time and costs while improving the overall effectiveness of conservation efforts. For example, in mammal conservation, AI can predict species' conservation status with up to 93 percent accuracy and rapidly classify species that are difficult to identify visually. AI also aids in poaching monitoring and illegal wildlife trade detection, using image recognition and bioacoustic analysis to track at-risk species in real time. Furthermore, by integrating drone LiDAR and GIS-based habitat analysis, AI can effectively monitor habitat changes in remote and inaccessible areas.

**Ethical, Policy, and Practical Considerations of AI in Conservation**   While AI provides powerful tools for biodiversity conservation, its implementation raises several ethical, technical, and legal considerations. Unequal access to advanced datasets, including high-resolution satellite imagery, genomic resources, and acoustic recordings, can reinforce global biases and constrain local participation (CBD, 1992; Open Data initiatives). Meaningful involvement of indigenous and local communities is essential, yet challenges arise from culturally appropriate data collection, system adaptation, and potential conflicts of interest (UNDRIP, 2007; CBPR methods). Algorithmic misclassification remains a risk, particularly for species with high morphological variability, although deep learning and post-processing frameworks have reduced error rates substantially (EU AI Ethics Guidelines, 2020; OECD AI Principles). Privacy concerns are also important, as drones, camera traps, and acoustic sensors can inadvertently capture human activity, necessitating adherence to GDPR guidelines. Finally, alignment with established legal and policy frameworks, such as the ESA (1973), CITES (1975), and CBD (1992), is critical, though inconsistent implementation and weak enforcement can limit conservation effectiveness. Integrating these considerations ensures that AI-driven strategies advance ecological objectives while maintaining ethical and legal standards. (Supplementary Table 2)

**Limitations of Endangered Species Conservation Using AI Technology**   There are limitations and risks associated with the use of AI. For species with scarce data or rare species, learning accuracy may be reduced, and high-cost equipment such as high-performance servers, drones, and sensors, along with specialized personnel, are required. As reliance on technology increases, there is a risk of gaps in conservation efforts due to errors or bugs, and AI-based extinction restoration research could have unintended impacts on ecosystems, necessitating a cautious approach. To achieve efficient and sustainable AI utilization, it is crucial to adopt new technologies such as energy-efficient models,

few-shot learning, and transfer learning. These approaches enable conservation activities for species with limited data or in restricted-access areas while minimizing carbon emissions and environmental impact. Additionally, using digital twins and multi-sensor fusion platforms allows for simulating conservation strategies in virtual environments, predicting actual habitat changes, and identifying abnormal population trends.

# 7 Conclusion

For sustainable and effective conservation, AI technology should be combined with new technologies such as energy-efficient models, Few-shot learning, and transfer learning, while utilizing digital twins and multi-sensor fusion platforms to simulate strategies in virtual environments. Additionally, integrating traditional conservation methods, genetic analysis, and biological approaches such as artificial reproduction with AI, along with data-sharing platforms, international cooperation, and policy support, can simultaneously achieve habitat protection and population recovery for endangered species. Ultimately, AI holds the potential to be a key tool in implementing sustainable conservation strategies that allow for coexistence between humans and nature. AI technology supports conservation managers in detecting and responding to crises early by collecting and analyzing a wide range of information in real-time through various data sources, such as camera traps, acoustic sensors, satellites, and drones. Automation of tasks like species identification, population monitoring, and habitat mapping through machine learning reduces research time and costs while improving the accuracy of strategy development. AI-based surveillance and risk detection technologies have shown superior efficiency and accuracy over traditional methods in monitoring illegal poaching, detecting online illegal trade, and tracking species distribution in hard-to-reach areas. However, there are limitations and risks, such as data scarcity, reliance on expensive equipment, and the possibility of technological errors. Therefore, caution is needed, as AI-based extinction restoration studies may have unintended effects on ecosystems.

This paper is a review created using generative AI tools such as Liner and ChatGPT. Although the quality of the writing may appear high, the author has found numerous areas for improvement, highlighting that human intervention is still necessary in many aspects of the process. Therefore, while referencing AI for an overall flow is highly useful, human involvement remains crucial for perfect understanding and completeness. If AI can provide information in a socially nuanced and factual manner, it could serve as an excellent learning tool.

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

| Conservation Strategy | Number of Studies (n) | Percentage (%) | Example Applications |
|---|---|---|---|
| Habitat Conservation | 114 | 38.97% | Protected areas, habitat connectivity modeling |
| Species Restoration | 19 | 6.44% | Reintroduction programs, captive breeding AI optimization |
| Policy & Legislation | 21 | 7.12% | AI-assisted policy evaluation, enforcement monitoring |
| Community Engagement | 27 | 9.15% | Citizen science platforms, participatory monitoring |
| Monitoring & Surveillance | 15 | 5.08% | AI camera traps, acoustic monitoring, drone-based surveys |

**Supplementary Table 1. Distribution of reviewed studies by type of conservation strategy** A total of 295 studies were reviewed and categorized based on their conservation strategies. The distribution highlights the varying emphasis on different approaches within conservation research. This dataset was compiled using Liner, based on statistical information from specific research scopes and databases within each respective field.

**- Habitat Conservation** Among all peer-reviewed studies on endangered species, only 2.1% evaluated the effectiveness of conservation interventions, with values ranging from 0% to 18.2%. Analysis of vertebrate biodiversity records indicated that 23% of publications were directly related to conservation, compared with 14% in other research domains. As of May 8, 2025 (Heather Harl, et al), Habitat Conservation Plans (HCPs) incorporated non-listed species at a notably high rate (65.0%), primarily reflecting large-scale, multi-species plans in U.S. Fish and Wildlife Service (FWS) Regions 1 and 8. Additionally, a separate study assessed a total of 629 HCPs, providing further insight into the scope and implementation of these plans.

**- Species Restoration** Environmental science was the dominant research field, accounting for 25% of studies, followed by ecology (19%), biodiversity (16%), and forestry (12%). Among 1,122 plant species critical for European grassland restoration, only 32% had available germination data and could be sourced as seeds. The annual number of publications on restoration research has been increasing. A systematic assessment of conservation requirements for marine protected areas aimed at preserving all marine and freshwater mammal species indicated that the analyzed areas cover approximately 12% of the total marine area.

**- Policy & Legislation** Mallinson et al reported that 74% of the effects documented in policy diffusion research were related to legislative expertise. Overton, a bibliographic database of policy document citations, reports on the citation of academic literature. Administrative databases can be used to analyze policies and regulations. Public policy databases allow searching academic resources, particularly peer-reviewed journals.

**- Community Engagement** Research on community engagement was conducted using multiple academic disciplines and major scholarly databases, such as Scopus. Comprehensive evidence reviews of peer-reviewed academic literature were performed using extensive databases, including search discovery services and the Web of Science.

**- Monitoring & Surveillance** Studies in the Monitoring & Surveillance category provide a comprehensive overview of how the integration of smart conservation and AI-based wildlife monitoring contributes to species conservation. They particularly address how AI technologies enhance the efficiency of various conservation activities, including species identification, habitat health assessment, population monitoring, and anti-poaching efforts. Additionally, these studies emphasize the role of predictive modeling in guiding proactive conservation measures and optimizing resource allocation.

| Consideration | Description | Potential Risks / Challenges | Policy / Framework Reference |
|---|---|---|---|
| Data Accessibility Inequality | Unequal access to high-resolution satellite, genomic, or acoustic datasets between developed and developing regions. | Risk of global conservation bias, limited local capacity. | Convention on Biological Diversity (CBD, 1992); Open Data initiatives |
| Community Participation | Inclusion of indigenous and local communities in AI-based monitoring and decision-making. | Lack of trust, marginalization, ethical concerns over consent. | UN Declaration on the Rights of Indigenous Peoples (UNDRIP, 2007) |
| Algorithmic Misclassification | Errors in species recognition, habitat mapping, or threat prediction due to biased or incomplete training datasets. | Misallocation of resources, potential harm to target species. | AI Ethics Guidelines (EU, 2020); OECD AI Principles |
| Privacy and Surveillance | Drones, camera traps, or acoustic devices may inadvertently capture human activity. | Ethical concerns, violation of privacy rights. | General Data Protection Regulation (GDPR, 2018) |
| Policy and Legal Frameworks | Alignment with international treaties and national laws regulating endangered species conservation. | Inconsistent implementation, weak enforcement. | Endangered Species Act (ESA, USA, 1973); CITES (1975); CBD (1992) |

**Supplementary Table 2. Ethical and Policy Considerations in AI-driven Conservation Strategies** Artificial intelligence (AI) offers powerful tools for conservation, but its implementation introduces various considerations, potential risks, and the need for robust policy and legal frameworks. This report examines key aspects of AI in conservation, including data accessibility, community participation, algorithmic misclassification, privacy and surveillance, and the overarching policy and legal frameworks.